# Memorization Capacity of Neural Networks with Conditional Computation

**Erdem Koyuncu**
Department of Electrical and Computer Engineering
University of Illinois Chicago
ekoyuncu@uic.edu

## Abstract

Many empirical studies have demonstrated the performance benefits of conditional computation in neural networks, including reduced inference time and power consumption. We study the fundamental limits of neural conditional computation from the perspective of memorization capacity. For Rectified Linear Unit (ReLU) networks without conditional computation, it is known that memorizing a collection of $n$ input-output relationships can be accomplished via a neural network with $O(\sqrt{n})$ neurons. Calculating the output of this neural network can be accomplished using $O(\sqrt{n})$ elementary arithmetic operations of additions, multiplications and comparisons for each input. Using a conditional ReLU network, we show that the same task can be accomplished using only $O(\log n)$ operations per input. This represents an almost exponential improvement as compared to networks without conditional computation. We also show that the $\Theta(\log n)$ rate is the best possible. Our achievability result utilizes a general methodology to synthesize a conditional network out of an unconditional network in a computationally-efficient manner, bridging the gap between unconditional and conditional architectures.

## 1 Introduction

### 1.1 Conditional Computation

Conditional computation refers to utilizing only certain parts of a neural network, in an input-adaptive fashion (Davis & Arel, 2013; Bengio et al., 2013; Eigen et al., 2013). This can be done through gating mechanisms combined with a tree-structured network, as in the case of "conditional networks" (Ioannou et al., 2016) or neural trees and forests (Tanno et al., 2019; Yang et al., 2018; Kontschieder et al., 2015). Specifically, depending on the inputs or some features extracted from the inputs, a gate can choose its output sub-neural networks that will further process the gate's input features. Another family of conditional computation methods are the so-called early-exit architectures (Teerapittayanon et al., 2016; Kaya et al., 2019; Gormez et al., 2022). In this case, one typically places classifiers at intermediate layers of a large network. This makes it possible to exit at a certain layer to reach a final verdict on classification, if the corresponding classifier is confident enough of its decision.

Several other sub-techniques of conditional computation exist and have been well-studied, including layer skipping (Graves, 2016), channel skipping in convolutional neural networks (Gao et al., 2019), or reinforcement learning methods for input-dependent dropout policies (Bengio et al., 2015). Although there are many diverse methods (Han et al., 2021), the general intuitions as to why conditional computation can improve the performance of neural networks remain the same: First, the computation units are chosen in an adaptive manner to process the features that are particular to the given input pattern. For example, a cat image is ideally processed by only "neurons that are specialized to cats." Second, one allocates just enough computation units to a given input, avoiding a waste of resources. The end result is various benefits relative to a network without conditional computation, including reduced computation time, and power/energy consumption (Kim & Seo, 2020). Achieving these benefits are especially critical in edge networks with resource-limited devices (Li et al., 2021a;b). Moreover, conditioning incurs minimal loss, or in some cases, no loss in learning performance.

Numerous empirical studies have demonstrated the benefits of conditional computation in many different settings. Understanding the fundamental limits of conditional computation in neural networks is thus crucial, but has not been well-investigated in the literature. There is a wide body of work on a theoretical analysis of decision tree learning (Maimon & Rokach, 2014), which can be considered as an instance of conditional computation. These results are, however, not directly applicable to neural networks. In Cho & Bengio (2014), a feature vector is multiplied by different weight matrices, depending on the significant bits of the feature vector, resulting in an exponential increase in the number of free parameters of the network (referred to as the capacity of the network in Cho & Bengio (2014)). On the other hand, the potential benefits of this scheme have not been formally analyzed.

## 1.2 Memorization Capacity

In this work, we consider the problem of neural conditional computation from the perspective of memorization capacity. Here, the capacity refers to the maximum number of input-output pairs of reasonably-general position that a neural network of a given size can learn. It is typically expressed as the minimum number of neurons or weights required for a given dataset of size, say $n$.

Early work on memorization capacity of neural networks include Baum (1988); Mitchison & Durbin (1989); Sontag (1990). In particular, Baum (1988) shows that, for thresholds networks, $O(n)$ neurons and weights are sufficient for memorization. This sufficiency result is later improved to $O(\sqrt{n})$ neurons and $O(n)$ weights by Vershynin (2020); Rajput et al. (2021). There are also several studies on other activation functions, especially the Rectified Linear Unit (ReLU), given its practicality and wide utilization in deep learning applications. Initial works Zhang et al. (2017); Hardt & Ma (2016) show that $O(n)$ neurons and weights are sufficient for memorization in the case of ReLU networks. This is improved to $O(\sqrt{n})$ weights and $O(n)$ neurons in Yun et al. (2019). In addition, Park et al. (2021) proves the sufficiency of $O(n^{2/3})$ weights and neurons, and finally, Vardi et al. (2022) shows that memorization can be achieved with only $O(\sqrt{n})$ weights and neurons, up to logarithmic factors. For the sigmoid activation function, it is known that $O(\sqrt{n})$ neurons and $O(n)$ weights (Huang, 2003), or $O(n^{2/3})$ weights and neurons (Park et al., 2021) are sufficient. Memorization and expressivity have also been studied in the context of specific network architectures such as convolutional neural networks (Cohen & Shashua, 2016; Nguyen & Hein, 2018).

The aforementioned achievability results have also been proven to be tight in certain cases. A very useful tool in this context is the Vapnik-Chervonenkis (VC) dimension (Vapnik & Chervonenkis, 2015). In fact, applying the VC dimension theory to neural networks (Anthony et al., 1999), it can be shown that the number of neurons and weights should be both polynomial in the size of the dataset for successful memorization. Specifically, $\Omega(\sqrt{n})$ weights and $\Omega(n^{1/4})$ neurons are optimal for ReLU networks, up to logarithmic factors. We will justify this statement later on for completeness.

## 1.3 Scope, Main Results, and Organization

We analyze the memorization capacity of neural networks with conditional computation. We describe our neural network and the associated computational complexity models in Section 2. We describe a general method to synthesize conditional networks from unconditional networks in Section 3. We provide our main achievability and converse results for memorization capacity in Sections 4 and 5, respectively. We draw our main conclusions in Section 6. Some of the technical proofs are provided in the supplemental material.

We note that this paper is specifically on analyzing the theoretical limits on neural conditional computation. In particular, we show that $n$ input-output relationships can be memorized using a conditional network that needs only $O(\log n)$ operations per input or inference step. The best unconditional architecture requires $O(\sqrt{n})$ operations for the same task. This suggests that conditional models can offer significant time/energy savings as compared to unconditional architectures. In general, understanding the memorization capacity of neural networks is a well-studied problem of fundamental importance and is related to the expressive power of neural networks. A related but separate problem is generalization, i.e. how to design conditional networks that can not only recall the memory patterns with reasonable accuracy but also generalize to unseen examples. The "double-descent" phenomenon (Belkin et al., 2019; Nakkiran et al., 2021) suggests that the goals of memorization and generalization are not contradictory and that a memorizing network can potentially also generalize well. A further

investigation of this phenomenon in the context of conditional networks, and the design of conditional networks for practical datasets remains beyond the scope of the present work.

**Notation:** Unless specified otherwise, all vector variables are column vectors. We use ordinary font (such as $u$) for vectors, and the distinction between a vector and scalar will be clear from the context. The symbols $O, \Omega$, and $\Theta$ are the standard Bachmann–Landau symbols. Specifically, $f_n \in O(g_n)$ means there is a constant $C > 0$ such that $f_n \leq Cg_n$ for sufficiently large $n$. On the other hand, if $f_n \in \Omega(g_n)$, then there is a constant $C > 0$ such that $f_n > Cg_n$ for sufficiently large $n$. We write $f_n \in \Theta(g_n)$ if $f_n \in O(g_n)$ and $f_n \in \Omega(g_n)$. The set $\mathbb{R}^p$ is the set of all $p$-dimensional real-valued column vectors. The superscript $(\cdot)^T$ is the matrix transpose. The function $\mathbf{1}(\cdot)$ is the indicator function, and $\lceil \cdot \rceil$ is the ceiling operator. A function $f(x)$ of variable $x$ is alternatively expressed as the mapping $x \mapsto f(x)$. Operator $\text{rank}(\cdot)$ is the rank of a matrix. Finally, $\| \cdot \|$ is the Euclidean norm.

## 2 SYSTEM MODEL

### 2.1 UNCONDITIONAL FEEDFORWARD NETWORKS

Consider an ordinary unconditional feedforward network of a homogeneous set of neurons, all of which have the same functionality. We also allow skip connections, which are mainly utilized for the generality of the model for converse results (The construction in our achievability results also skip layers, but at most one layer at a time). Formally, we consider an $L$-layer network with the ReLU operation $\phi(x) = \max\{0, x\}$, and the input-output relationships

$$y_\ell = \phi\left(W_\ell \left[y_{\ell-1}^T \cdots y_0^T\right]^T\right), \ell = 1, \ldots, L, \tag{1}$$

where $y_0$ is the input to the neural network, $y_\ell$ is the output at Layer $\ell$, and $y_L$ is the neural network output. Also, $W_\ell$ is the weight matrix at Layer $\ell$ of appropriate dimensions.

### 2.2 MEASURING THE COST OF COMPUTATION

A recurring theme of the paper will be to calculate the output of a neural network given an arbitrary input, however with low computational complexity. In particular, in (1), given the outputs of Layer $\ell - 1$, the calculation of the outputs of Layer $\ell$ can be accomplished through multiplication with matrix $W_\ell$, followed by the activation functions. Each ReLU activation function can be implemented via a simple comparison given local fields. Hence, calculation of the output of Layer $\ell$ can be accomplished with $O(\dim W_\ell)$ operations (multiplications, additions, and comparisons), and the output $y_L$ of the entire network can be calculated using $O(\sum_\ell \dim W_\ell)$ operations. Here, $\dim W_\ell$ represents the product of the number of rows and columns in $W_\ell$. In other words, $\dim W_\ell$ is the number of dimensions in $W_\ell$. Hence, in unconditional architectures, the number of operations to calculate the output of the network is essentially the same (up to constant multipliers) as the number of weights in the network. With regards to our basic measure of complexity, which relies on counting the number of operations, one can argue that multiplication is more complex than addition or comparison, and hence should be assigned a larger complexity. We hasten to note that the relative difficulty of different operations will not affect our final results, which will have an asymptotic nature.

It is instructive to combine the $O(\sum_\ell \dim W_\ell)$ complexity baseline with the memorization results described in Section 1. Let $X = \{x_1, \ldots, x_n\} \subset \mathbb{R}^p$ be a dataset of inputs. Let $d_1, \ldots, d_n \in \mathbb{R}^r$ be the corresponding desired outputs. In the memorization task, one wishes to design a network that can provide an output of $d_i$ for an input of $x_i$ for each $i \in \{1, \ldots, n\}$. It is known that, up to logarithmic factors, $O(\sqrt{n})$ weights and neurons are sufficient for memorization of $n$ patterns (Vardi et al., 2022). It follows from the discussion in the paragraph above that $O(\sqrt{n})$ operations sufficient to recall a stored memory pattern (i.e. to calculate the output of the neural network for a given $x_i$). The goal of this paper is to accomplish the same task using much fewer operations required per input. Specifically, we will show how to do perfect recall using only $O(\log n)$ operations per input. We shall later show that $\Theta(\log n)$ is, in fact, the best possible rate.

### 2.3 CONDITIONAL NETWORKS

In order to achieve perfect recall using a subpolynomial number of operations, we use the idea of conditional computation. The conditioning model that we utilize in this work is a simple but

general scenario where we allow executing different sub-neural networks depending on how an intermediate output of the network compares to some real number. Each conditioning is thus counted as one operation. Formally, we describe a conditional neural network via a rooted full binary tree where every vertex has either 0 or 2 children. Every vertex is a sequence of operations of the form $\mathrm{v}_{n+1} \leftarrow \phi(\beta_n \mathrm{v}_n + \cdots + \beta_1 \mathrm{v}_1)$, where $\beta_1, \ldots, \beta_n$ are weights, and variables $\mathrm{v}_1, \ldots, \mathrm{v}_n$ are either (i) inputs to the neural network, or (ii) defined as new variables preceding $\mathrm{v}_{n+1}$ at the same vertex or at one of the ancestor vertices. Every edge is a conditioning $\mathrm{v} \circ \beta$, where $\mathrm{v}$ should be defined at any one of the vertices that connects the edge to the root node, $\beta$ is a constant weight, and $\circ \in \{\leq, <, =, \neq, >, \geq\}$. We assume that the two edges connected to the same vertex correspond to complementary conditions; e.g. $\mathrm{v}_1 < 3$ and $\mathrm{v}_1 \geq 3$.

An example conditional network, expressed in algorithmic form, is provided in Algorithm 1, where $u$ represents the input, $o$ is the output, and the subscripts are the vector components. For example, if the input is a vector $u$ with $u_1 > 3$, resulting in an intermediate feature vector with $z_3 \neq 5$, then two operations are incurred due to the conditionings in Lines 1 and 5, and $O(\dim(W_1) + \dim(W_2) + \dim(W_5))$ operations are accumulated due to neural computations.

Our model encompasses various neural conditional computation models in the literature. One example is early exit architectures (Teerapittayanon et al., 2016; Kaya et al., 2019; Gormez & Koyuncu, 2022). As described in Section 1, a typical scenario is where one places intermediate classifiers to a deep neural network. If an intermediate classifier is confident enough of a decision, then an "early exit" is performed with the corresponding class decision. Here, one skips subsequent layers, saving computation resources. The decision to exit is often a simple threshold check, e.g. whether one of the soft probability outputs of the intermediate classifier exceeds a certain threshold. Hence, most early exit networks can be modeled via the simple if-else architecture described above. Mixture of experts (MoE) architectures Shazeer et al. (2017); Fedus et al. (2022) can also be realized under our model. In this case, there are multiple gating networks, each of which is responsible for one expert. One routes a feature vector to only a subset of experts whose gating networks have the largest outputs. The choice of experts can be accomplished through if-else statements. For example, for two gates and experts, the gate with the largest output can be found by comparing the difference between the two gates' outputs against zero. Another gating approach that can be realized as a special case of our model can be found in Cho & Bengio (2014).

**Algorithm 1** An example conditional neural network

1: **if** $u_1 > 3$ **then**
2: $\quad z = \phi(W_2 \phi(W_1 u))$
3: $\quad$ **if** $z_3 = 5$ **then**
4: $\quad\quad o = \phi(W_4 z)$
5: $\quad$ **else**
6: $\quad\quad o = \phi(W_5 z)$
7: $\quad$ **end if**
8: **else**
9: $\quad o = \phi(W_3 u)$
10: **end if**

## 3 SYNTHESIZING A CONDITIONAL NETWORK OUT OF AN UNCONDITIONAL NETWORK

Consider an arbitrary unconditional network as in (1), whose implementation requires $O(\sum_\ell \dim W_\ell)$ operations, as discussed in Section 2.2. Suppose that the network is well-trained in the sense that it can already provide the output $d_i$ for a given input $x_i$, for every $i \in \{1, \ldots, n\}$. Out of such an unconditional network, we describe here a general methodology to synthesize a conditional network that requires much fewer than $O(\sum_\ell \dim W_\ell)$ operations, while keeping the input-output relationships $x_i \mapsto d_i$, $i \in \{1, \ldots, n\}$ intact.

We first recall some standard terminology (Haykin, 2008). Consider the neuron $[x_1, \ldots, x_n] \mapsto \phi(\sum_i x_i w_i)$. We refer to $x_1, \ldots, x_n$ as the neuron inputs, and $w_1, \ldots, w_n$ as the neuron weights. We can now provide the following definition.

**Definition 1.** *Suppose that the activation function satisfies $\phi(0) = 0$. Given some fixed input to a neural network, a neuron with at least one non-zero input is called an active neuron. A neuron that is not active is called an inactive or a deactivated neuron. A weight is called active if it connects an active neuron to another active neuron and is non-zero. A weight is called inactive if it is not active.*

The source of the phrase "inactive" is the following observation: Consider an input to the neural network and the corresponding output. By definition, we will obtain the same output after removing all inactive neurons from the network for the same input. We can simply "ignore" inactive neurons. Likewise, we can remove any inactive weight and obtain the same output.

Our idea is to condition the computation on the set of active neurons and the corresponding active weights. Note that removing the inactive weights and neurons does not change the network output. Moreover, often the number of active weights given an input can be significantly lower than the overall number of neurons of the unconditional architecture (which determines the number of operations required to perform inference on the unconditional network), resulting in huge computational savings. The complication in this context is that the set of inactive weights depends on the network input. Fortunately, it turns out that determining the set of active weights can be accomplished with fewer operations than actually computing the local fields or outputs of the corresponding active neurons. The final result is provided by the following theorem.

**Theorem 1.** *Consider an arbitrary dataset $X = \{x_1, \ldots, x_n\} \subset \mathbb{R}^p$ of inputs. Let $d_1, \ldots, d_n \in \mathbb{R}^r$ be the corresponding desired outputs. Suppose that the unconditional neural network defined by (1) satisfies the desired input-output relationships in the sense that for any $i$, if the input to the network is $x_i$, then the output is $d_i$. Also suppose that the input $x_i$ results in $\omega_i$ active weights. Then, there is a conditional network that similarly satisfies all desired input output relationships, and for every $i$, performs at most $p + 4\omega_i$ operations given input $x_i$.*

*Proof.* (Sketch) By an " input configuration," we mean a certain subset of neurons that are active at a certain layer of the unconditional "base" network. We represent input configurations by binary vectors where "0" represents an inactive neuron, while "1" represents an active neuron. As an example, the input configuration $[1\,0\,1]^T$ refers to a scenario where only the first and the third neurons at the layer is active. Analogous to input configurations, "output configurations" are binary vectors that represent whether neurons provide zero or non-zero outputs. For example, an output configuration of $[1\,1\,0]^T$ means that only the first and the second neurons provide a non-zero output.

Consider first an unconditional network without skip connections. The key idea is to observe that, given the output configuration of Layer $\ell - 1$, one can uniquely obtain the input configuration of Layer $\ell$. Hence, a conditional network can be designed to operate in the following manner: Given an input, we first find the output configuration $\texttt{OC}_0$ at Layer $0$, which is nothing but the non-zero components of the input vector. This can be done by $p$ conditioning statements, where $p$ is the input dimension. The result is a unique input configuration $\texttt{IC}_1$ at Layer $1$. Meanwhile, we can obtain the output $y_1$ of Layer $1$ by only calculating the outputs of neurons that correspond to the non-zero components of $\texttt{IC}_1$, since other neurons at Layer $1$ are guaranteed to have all-zero inputs and thus provide zero output. This can be accomplished via $O(a_1)$ multiplications and additions, where $a_\ell$ represents the number of active weights in Layer $\ell$. Then, we find the output configuration $\texttt{OC}_1$ at Layer $1$, using $|\texttt{IC}_1|$ conditioning statements on $y_1$. Having obtained $\texttt{OC}_1$, we can similarly find the Layer $2$ input configuration and output. The conditional network processes the remaining layers recursively in the same manner. The functionality of the unconditional network is reproduced exactly so all input-output relationships are satisfied. Given that $\sum_\ell a_\ell = \omega_i$, the total complexity is $p + O(\omega_i)$. We refer to the complete proof in Appendix A for the precise bound and generalization to networks with skip connections. $\qquad\qquad\square$

The proof of the theorem in Appendix A suggests that the true complexity is closer to $2\omega_i$ than the actual formal upper bound provided in the theorem statement. The number $2\omega_i$ stems from the $\omega_i$ additions and $\omega_i$ multiplications that are necessary to calculate the local fields of active neurons.

Theorem 1 shows that a design criterion to come up with low computational-complexity neural networks might be to put a constraint on the number of active weights given any training input to an ordinary, unconditional feedforward architecture. Using the theorem, we can synthesize a conditional neural network with the same functionality as the unconditional feedforward network. In the next section, we will apply this idea to the problem of memorization capacity.

We conclude this section by noting that Theorem 1 may prove to be useful in other applications. For example, any method that results in sparse representations or weights, such as pruning, will result in many inactive neurons and weights. Sparsity is useful; however, unstructured sparsity is difficult to exploit: For example, multiplying by a matrix half of whose entries are zero at random positions will likely be as difficult as multiplying by a random matrix without sparsity constraints. The construction in Theorem 1 may provide computational savings for such scenarios.

Another observation is how Theorem 1 can potentially simplify the design of conditional networks. In any conditional architecture, an important question is where to place the decision gates that route

information to different parts of the network. It is a combinatorially very difficult if not impossible to optimize a heterogeneous collection of "ordinary neurons" and "gates." Such an optimization is also completely unsuitable for a gradient approach. Hence, most previous works provide some empirical guidelines for gate placement, and do not optimize over the gate locations once they are placed according to these guidelines. The message of Theorem 1 is that there is no need to distinguish between gates and neurons. All that needs to be done is to train an unconditional network consisting only of ordinary neurons to be as "inactive" as possible, e.g. by increased weight sparsity through regularization Louizos et al. (2018). The theorem can then construct a conditional network that can exploit the sparsity to the fullest, placing the gates automatically.

## 4 MEMORIZATION WITH LOW COMPUTATIONAL COMPLEXITY

In this section, we introduce our ReLU network to memorize the input-output relationships $x_i \mapsto d_i$, $i = 1, \ldots, n$. Our network will only yield $O(\log n)$ active neurons and weights per input. Application of Theorem 1 will then imply a conditional network that can achieve perfect recall with only $O(\log n)$ operations per input, as desired.

Given a neural network $f$, and an input $x$ to $f$, let $A(x; f)$ and $W(x; f)$ denote the number of active neurons and active weights, respectively. Our feedforward ReLU network construction is provided by the following theorem.

**Theorem 2.** *Let $X = \{x_1, \ldots, x_n\} \subset \mathbb{R}^p$ be a dataset of input patterns such that for every $i, j \in \{1, \ldots, n\}$ with $i \neq j$, we have $x_i \neq x_j$. Let $\bar{x}_i = \left[\begin{smallmatrix} 1 \\ x_i \end{smallmatrix}\right] \in \mathbb{R}^q$, where $q = 1 + p$, be the augmented dataset patterns for biasing purposes. Let $d_1, \ldots, d_n \in \mathbb{R}^r$ be the corresponding desired outputs. Suppose every component of $d_i$ is non-negative for each $i \in \{1, \ldots, n\}$. Then, there is a neural network $f : \mathbb{R}^q \to \mathbb{R}^r$ such that for every $i \in \{1, \ldots, n\}$, we have $f(\bar{x}_i) = d_i$,*

$$A(\bar{x}_i; f) \leq 2(q+1)\lceil \log_2 n \rceil + q + r \in O(r + q \log n), \tag{2}$$

$$W(\bar{x}_i; f) \leq 12q(q+1)\lceil \log_2 n \rceil + (r+2)q + r - 2 \in O(rq + q^2 \log n). \tag{3}$$

*Proof.* (Sketch) We provide an illustrative sketch of the proof. Suppose we wish to memorize the two-dimensional dataset given by Fig. 1a. We can divide the datasets to two parts via a line, the resulting two parts to further two sub-parts, and so on, until reaching singletons, as shown in Fig. 1b. The overall network architecture that achieves the performance in the statement of the theorem is then shown in Fig. 2. The initial block $\mathbf{T}$ is a basic preprocessing translation. The "switch" $\mathbf{S}_{ij}$ corresponds to the line parameterized by weights $w_{ij}$ in Fig. 1b. The switch $\mathbf{S}_{ij}$ routes the zero vector to one output path and copy its input to the other output path, depending on the side of the $w_{ij}$-line its input vector resides. The switches are followed by ReLU neurons with weights $\gamma_i$. These neurons map the corresponding input pattern on the active path to its desired output. Finally, the output of the $\gamma_i$-neurons are accumulated. As an example, the signals on the graph for input pattern $\bar{x}_6$ is provided, with $\bar{\bar{x}}_6 \triangleq \mathbf{T}(\bar{x}_6)$. In general, all input-output relationships are satisfied with $O(\log n)$ active neurons per dataset sample. We refer to the complete proof in Appendix B for the precise bounds. $\square$

**Corollary 1.** *Let $x_i, d_i$, $i = 1, \ldots, n$ be a sequence of arbitrary input-output pairs as stated in Theorem 2. Then, there is a conditional network that, for every $i$, provides an output of $d_i$ whenever the input is $x_i$ by performing only $O(rq + q^2 \log n)$ operations.*

*Proof.* We apply the synthesis procedure in Theorem 1 to the network constructed in Theorem 2. An alternative more "direct" proof (that does not need Theorem 1) is to implement the gating blocks $\mathbf{S}_{ij}$ in Fig. 1 via if-else conditioning statements, resulting in a similar architecture. $\square$

We recall that for ReLU networks without conditional computation, the best achievability result (Vardi et al., 2022) requires $O(\sqrt{n \log n})$ neurons and weights for memorizing a dataset of size $n$. Since the construction in Vardi et al. (2022) is not optimized for conditional computation, it is easily observed that every input activates all $O(\sqrt{n \log n})$ neurons and weights of the network, resulting in $O(\sqrt{n \log n})$ active weights per input. In fact, the remarkable construction in Vardi et al. (2022) is a narrow network of a finite width of only 12, but depth $O(\sqrt{n \log n})$. Even if every input activates only one neuron at each layer, one obtains $O(\sqrt{n \log n})$ active neurons or arithmetic operations per input. In contrast, we only need $O(\log n)$ active weights or operations (via Corollary 1) per input.

The fact that one cannot fundamentally achieve a sub-polynomial number of neurons without conditional computation is an easy consequence of the VC dimension theory. In our setting, the VC dimension corresponds to the cardinality of the largest dataset that can be memorized. Hence, upper bounds on the VC dimension translate to lower bounds on the memorization capacity. In this context, Anthony et al. (1999, Theorem 8.6) provides an $O(n_w^2)$ upper bound on the VC dimension for ReLU networks, where $n_w$ denotes the number of weights in the network. Since any upper bound on the VC dimension is also an upper bound on the cardinality of the largest possible dataset that can be memorized, we have $n \in O(n_w^2)$. It follows that $n_w \in \Omega(\sqrt{n})$ weights are the best possible for ReLU networks without conditional computation, meaning that the results of Vardi et al. (2022) are optimal up to logarithmic factors. Also, using the bound $n_w \leq n_e^2$, where $n_e$ is the number of neurons in the network (equality holds in the extreme scenario where every neuron is connected to every other neuron, including itself), we obtain the necessity of $\Omega(n^{1/4})$ neurons.

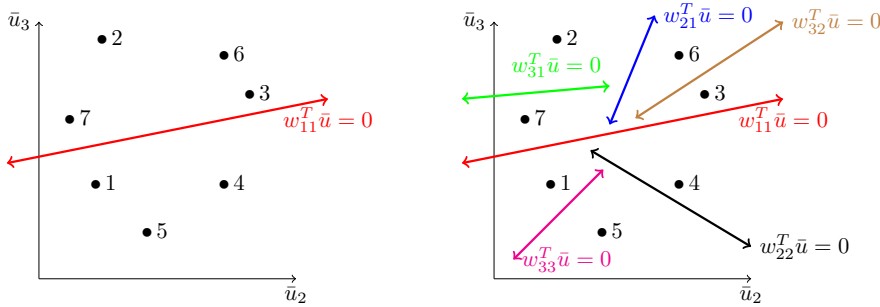

(a) Dividing a set of point to two equal subsets.   (b) Continued divisions until reaching singletons.

Figure 1: The divide and conquer strategy.

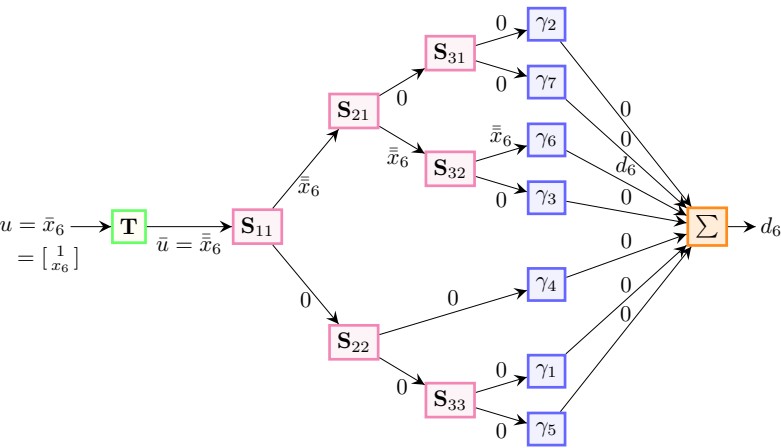

Figure 2: An example network architecture for the achievability result. The block $\mathbf{T}$ represents the transformation in Step 1. Blocks $\mathbf{S}_{ij}$ are the routing switches. Blocks $\gamma_i$ represent ReLU neurons with weights $\gamma_i$, and the $\sum$ block represents a ReLU neuron with all-one weights.

In contrast to the deep and narrow architecture that is optimal for unconditional computation, the network that achieves the performance in Theorem 2 is considerably wider but much shallower. In fact, the proof in Appendix B reveals that our network has width $O(n)$, with $O(n)$ nodes, and depth $O(\log n)$. These numbers are a consequence of the classical binary decision tree that we utilize in our network construction. Thanks to conditional computation, although the network has $O(n)$ nodes, every dataset pattern activates only $O(\log n)$ neurons instead of $O(\sqrt{n})$ neurons without conditional computation. Note that the function $\log n$ grows much slower than $\sqrt{n}$ so that the number of neurons activated by the conditional computation is asymptotically negligible relative to an unconditional network. Moreover, every neuron in the conditional computation network has a bounded number of $q$ weights. The two facts translate to big savings in terms of the cost of computation, thanks to Theorem 1. An interesting open problem that we shall leave as future work is whether one can achieve the

same performance with a much smaller network size, e.g. with $O(\sqrt{n})$ neurons, which is known to be optimal. This will also help reduce the size of the network synthesized by Theorem 1.

It should be mentioned that the bounds (2) and (3) on the number of active neurons and weights as stated in Theorem 2 holds only for input patterns that belong to the dataset. For such patterns, only one path from the input to the output of the neural network is activated. The global behavior of the neural network for arbitrary inputs is more complex. A careful analysis of the construction in Appendix B reveals that near the decision boundaries (the lines in Fig. 1b), multiple paths of the network are activated, This will result in more active neurons and weights than what is suggested by the upper bounds in (2) and (3), respectively. However, the measure of such pathological inputs can be made arbitrarily small by tuning the switches appropriately.

## 5  ULTIMATE COMPUTATIONAL LIMITS TO MEMORY RECALL

In the previous section, we showed that $O(\log n)$ operations is sufficient to perfectly recall one of $n$ input-output relationships. We now analyze the necessary number of operations per input for successful recall. Our main result in this context is provided by the following theorem.

**Theorem 3.** *Let the input vectors $x_1, \ldots, x_n \in \mathbb{R}^p$ and the corresponding desired output vectors $d_1, \ldots, d_n \in \mathbb{R}$ satisfy the following property:*

- *The matrix $\begin{bmatrix} x_{i_1} & \cdots & x_{i_{p+1}} \\ d_{i_1} & \cdots & d_{i_{p+1}} \end{bmatrix}$ has rank $p+1$ for any subset $\{i_1, \ldots, i_{p+1}\}$ of $\{1, \ldots, n\}$.*

*Suppose that a conditional network $f$ satisfies the desired input-output relationships: For every $i$, the output of the network is $d_i$ whenever the input is $x_i$. Also, assume that the number of operations performed on each $x_i$ is bounded above by some $\alpha \geq 0$. Then, we have $\alpha \geq \log_2 \frac{n}{p} \in O(\log n)$.*

*Proof.* Since there are at most $\alpha$ operations per input, there are at most $\alpha$ comparisons per input as well, counting the comparisons needed to implement the neuron activation functions. We can represent the entire conditional computation graph/network as a binary tree where the distance between the root and a leaf node is at most $\alpha$. This results in tree of at most $2^\alpha$ leaf nodes. Each node of the tree compares real numbers to intermediate variables, which are linear functions of network inputs or other intermediate variables. Assume now the contrary to the statement of the theorem, i.e. the number of operations for every input is at most $\alpha$, all input-output relationships are satisfied, but $n > 2^\alpha p$. Since there are at most $2^\alpha$ leaf nodes, there is at least one leaf node that admits $1+p$ inputs (i.e. there are $1+p$ input patterns of the dataset such that the corresponding path over the tree ends at the leaf node). Without loss of generality, suppose the indices for these inputs are $\{1, \ldots, 1+p\}$. Writing down the input output relationship of the network for the leaf node, we obtain

$$[d_1 \cdots d_{1+p}] = W_0 [x_1 \cdots x_{1+p}] \tag{4}$$

for some $q \times p$ matrix $W_0$. This relationship follows, as by fixing a path on the tree, we obtain the unique linear transformation $W_0$ that maps the inputs to the neural network to its outputs. According to the Rouché–Capelli theorem (Shafarevich & Remizov, 2012, Theorem 2.38), a necessary condition for the existence of $W_0$ to solve (4) is

$$\text{rank}([x_1 \cdots x_{p+1}]) = \text{rank}\left( \begin{bmatrix} x_1 & \cdots & x_{p+1} \\ d_1 & \cdots & d_{p+1} \end{bmatrix} \right). \tag{5}$$

On the other hand, as a result of the rank condition stated in the theorem, the left hand side rank evaluates to $p$, while the right hand side evaluates to $1+p$. We arrive at a contradiction, which concludes the proof of the theorem. $\square$

The condition on the dataset and the desired outputs that appear in the statement of Theorem 3 is, up to a certain extent, necessary for the lower bound to hold. For example, if the desired outputs can simply be obtained as a linear transformation of inputs, then one only needs to perform a constant number of operations for each input to obtain the desired outputs, and the lower bound becomes invalid. In this context, the rank condition ensures that subsets of outputs cannot be obtained as linear functions of inputs. However, it should also be mentioned that the condition is not particularly restrictive in limiting the class of datasets where the theorem holds. For example, if the components

of the dataset members and the corresponding desired outputs are sampled in an independent and identically distributed manner over any continuous distribution with positive support, it can be shown that the rank condition will hold with probability 1. Hence, it can be argued that almost all datasets satisfy the rank condition and thus obey the converse result.

Corollary 1 has shown that $n$ patterns can be memorized using $O(\log n)$ operations per input. The matching $\Omega(\log n)$ lower bound in Theorem 3 proves that the $\Theta(\log n)$ rate is the best possible. However, the two results do not resolve how the number of operations should scale with respect to the input and output dimensions.[1] This aspect of the problem is left as future work.

## 6 CONCLUSIONS AND DISCUSSIONS

We have studied the fundamental limits to the memorization capacity of neural networks with conditional computation. First, we have described a general procedure to synthesize a conditional network out of an ordinary unconditional feedforward network. According to the procedure, the number of operations required to perform inference on an input in the synthesized conditional network becomes proportional to the number of so-called "active weights" of the unconditional network given the same input. This reduces the problem of designing good conditional networks to the problem of designing ordinary feedforward networks with a low number of active weights or nodes. Using this idea, we have shown that for ReLU networks, $\Theta(\log n)$ operations per input is necessary and sufficient for memorizing a dataset of $n$ patterns. An unconditional network requires $\Omega(\sqrt{n})$ operations to achieve the same performance. We also described a method to synthesize a conditional network out of an unconditional network in a computationally-efficient manner.

One direction for future work is to study the memorization capacity for a sum-constraint, as opposed to a per-input constraint on the number of operations. While a per-input constraint makes sense for delay-sensitive applications, the sum-constrained scenario is also relevant for early-exit architectures, where there is a lot of variation on the size of active components of the network. Extensions of our results to different activation functions or to networks with bounded bit complexity can also be considered. In this context, Vardi et al. (2022) shows that, for every $\epsilon \in [0, \frac{1}{2}]$, $\Theta(n^\epsilon)$ weights with $\Theta(n^{1-\epsilon})$ bit complexity is optimal for memorizing $n$ patterns, up to logarithmic factors. This result was proven under a mild separability condition, which restricts distinct dataset patterns to be $\delta$-separated in terms of Euclidean distance. The optimality of the results of Vardi et al. (2022) suggests that under a similar separability condition, the bit complexity of our designs can also potentially be reduced without loss of optimality. This will remain as another interesting direction for future work.

Many neural network architectures rely heavily on batch computation because matrix multiplications can be performed very efficiently on modern processors. In this context, one disadvantage of conditional architectures is their general misalignment with the idea of batching. Nevertheless, if there are not too many branches on the network, and if the branch loads are balanced, each subbranch can still receive a relatively large batch size. Fortunately, only a few gates can promise significant performance gains (Fedus et al., 2022). More work is needed, however, to make networks with aggressive conditioning more efficient in the batch setting. In this context, our synthesis theorem can potentially enable conditional networks to be trained as if they are unconditional, enabling batch computation. We note that in techniques like soft-gating (Shazeer et al., 2017), a batch also traverses the entire computation graph during training (Kaya et al., 2019).

The main goal of this paper has been to derive theoretical bounds on the memorization capacity. More research is clearly needed for practical methods to train neural networks that can effectively utilize conditional computation and also generalize well. We hope that the constructions and theoretical bounds provided in this paper will motivate further research in the area.

ACKNOWLEDGMENTS

This work was supported in part by National Science Foundation (NSF) under Grant CNS-2148182 and in part by Army Research Lab (ARL) under Grant W911NF-2120272.

---

[1] Although Theorem 3 only considers scalar desired outputs, it can easily be extended to the multi-dimensional case. In fact, a successful memorization of, say, two-dimensional output vectors with $o(\log n)$ active neurons would imply the successful memorization of scalar outputs with the same number of neurons (simply by ignoring the neurons that provide the second component of the output), contradicting Theorem 3.

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
