# OpenReview forum: "Memorization Capacity of Neural Networks with Conditional Computation"
_ICLR.cc/2023/Conference — ICLR 2023 poster_

### Official Review · Reviewer_4ix4 · 2022-10-23

**Confidence:** 4
**Correctness:** 3
**Technical Novelty And Significance:** 2
**Empirical Novelty And Significance:** Not applicable
**Recommendation:** 5

**Clarity, Quality, Novelty And Reproducibility:**

In Def 1, it seems that the authors defined "input" to a neuron to be each of the incoming directed edges. However, this confused me quite a bit because I thought the input to a neuron is just a single scalar, a weighted combination of the output of the previous layer. This made it difficult for me to understand "at least one non-zero input" because in my view there is only one input to a neuron if we have a fixed network input. Please consider trying to clarify the definition of "input" to a neuron.

While I got the main idea after quite a bit of time, the proof of Theorem 1 is very confusing to read. First of all, the definition of depth $\ell$ relative to layer $\ell$ is confusing. At the top of page 5 it is said "the nodes at depth $\ell$ correspond to all input configurations by … Layer $\ell$" but in Example 1, "the node 110 at Depth 1" and "an input configuration of 11 at Depth 2 (Layer 3)" makes me think that this may not be the case.

Also, the definition of root node is also confusing; again, the top of page 5 says "there is only one input configuration at depth 0, corresponding to the input layer" and this becomes the root node of the tree. So I thought the root node corresponds to the input layer. Then, two paragraphs later it is said "Each of the $2^{α_{11}}$ edges originating from the root node represents the $2^{α_{11}}$ possible output configurations at Layer 1", suggesting that the first hidden layer (Layer 1) actually is the root node.

All in all, while the proof delivers the main idea, I thought it might be helpful to provide concrete "algorithms" that construct a configuration tree from an unconditional network and then construct a conditional network from the configuration tree. The paper could use some polishing passes to clean up the confusing definitions too.


**Strength And Weaknesses:**

I have not seen any theoretical results on conditional neural networks and I believe this paper is one of the first to tackle theoretical problems for conditional networks; this is a plus. However (perhaps because I'm a theory person), I am not sure if this class of architecture involving explicit if-else branching is really widely used in practice.

The paper constructs a conditional ReLU network that recalls a memorized output in $O(\log n)$ operations, and shows that these many operations are actually necessary. Hence, this paper develops a tight characterization of the "operation complexity" for memorizing datasets with conditional networks.

However, my honest opinion is that both the sufficiency and necessity results are not significant enough to merit acceptance. For sufficiency results, one can easily prove the same statement by constructing a conditional network in the following way:
1) Since the $x_i$'s are all distinct, we can choose a vector $w$ such that the $n$ scalars $w^T x_1, \dots, w^T x_n$ are all distinct. This is a standard first step in many results on memorization.
2) Next, we can construct the "binary decision tree" that divides the $n$ data points to $n$ different leaves, which can be implemented with the if-else clauses of a conditional network. The tree can be constructed in a way that each data point $x_i$ goes through $\lceil \log_2 n \rceil$ conditioning operations.
3) At each leaf corresponding to $w^T x_i$, we choose appropriate network parameters to map $w^T x_i$ to $y_i$.

Of course, the paper does it in a different way, by proving a conversion theorem from unconditional to conditional (Theorem 1) and then proving a construction of an unconditional ReLU network (Theorem 2). However, the main idea is the same; to construct a binary decision tree that we can deal with each example separately. To me, Theorems 1 and 2 look like a detour to prove a simple thing in an unnecessarily more complicated way.

Also, for Theorem 3, the tightness of the construction is proven only in terms of $n$; rather counterintuitively, the lower bound $\Omega(\log \frac{n}{p})$ obtained in Theorem 3 decays with increasing input dimension $p$. This suggests that the theorem could be further improved to capture the right dependence on input/output dimension, as also noted by the authors.

Another weakness is that the paper is not cleanly written, especially the proof part. I have read all the proofs and got the main idea, but there are several clarity issues that make them confusing to read. More on that in the "Clarity" part below.

In terms of the literature review part, it looks to me that the paper cites most of the noteworthy papers on memorization capacity of neural networks, but I spotted two papers are missing.
- "Optimization landscape and expressivity of deep CNNs" establishes memorization results for CNNs.
- "Small ReLU networks are powerful memorizers: a tight analysis of memorization capacity" shows sufficiency of $O(\sqrt{n})$ neurons and $O(n)$ weights for ReLU networks.

Minor comments:
- When defining unconditional feedforward networks, the authors say that skip connections are "allowed for generality," and from Eq (1) we can see that the model is allowing skip connections of arbitrary length (i.e., denseNet-style jump connections). However, for expressive power results, I do not think that allowing/exploiting skip connections necessarily improves the generality of the theorem, because skip connections are usually quite useful for reducing the number of weights/neurons required to construct a certain network. I have checked Theorem 2 and it seems the results can actually hold without using skip connections, at the cost of slightly increased number of operations.

- The proof of Theorem 1 left me with three questions. First, why should all the $2^{\alpha_{i\ell}}$ output edges be drawn even if many of the arrows are left unused (i.e., does not make it all the way to the output layer/leaf node)? When you construct the conditional network, what do you do to the unused edges that do not extend all the way to the output layer? Lastly, although the theorem allows arbitrary skip connections, the construction of the configuration tree seems to only take into account the connections between adjacent layers. How do you deal with faraway skip connections?

- Lemma 1 seems to use $2q+2$ neurons instead of $q+2$? Also, the statement says "two-layer network" which is a synonym to "one-hidden-layer" to many authors, whereas the actual construction involves two hidden layers.

P2: A very useful too -> tool

P6: In Step-2, what is $d$? If you meant the input dimension it should be $p$.

Thm 2: every neuron in network $f$ has at most $q$ weights—add "incoming" before weights?

P7: $n_w$ and $n_e$ are both defined to be "the number of weights of the network," the difference is not at all clear


**Summary Of The Paper:**

This paper studies the memorization capacity of conditional ReLU networks. By a conditional network we mean a network that allows "branching" of the flow of computation by conditional expressions (i.e., if-else statements).

- The paper develops a general recipe that converts a general fully-connected unconditional neural network to a conditional network that only utilizes "active neurons" used for computing the output of a given input (Theorem 1).

- Based on that, the paper then constructs an unconditional ReLU network $f$ which memorizes $n$ arbitrary input-output pairs $(x_i, d_i)$ where each pair requires the use of only $O(\log n)$ "active neurons" to compute the correct output $d_i = f(x_i)$ (Theorem 2).

- Theorems 1 and 2 together yield Corollary 1, which establishes the existence of a conditional ReLU network which memorizes $n$ input-output pairs and computes the output $d_i = f(x_i)$ in $O(\log n)$ operations.

- Theorem 3 shows that $\Omega(\log n)$ operations are in fact necessary for memorizing and recalling $n$ points.

**Summary Of The Review:**

This paper studies memorization capacity of conditional neural networks. It develops a conditional ReLU network that can memorize a given dataset and compute the output of each data point in $O(\log n)$ operations, and this number of operations is tight in terms of $n$. While a tight characterization in $n$ is valuable, I honestly think that the results are not significant enough to merit acceptance. Also, the paper has several clarity issues which makes it difficult to read and understand the paper. Therefore, I recommend rejection.

---

> ### Author Response · Authors · 2022-11-18
> **Response to Reviewer 4ix4 (1 of 3)**
>
> (1 of 3)
>
> Dear Reviewer,
>
> Thank you very much for your constructive comments and the time and effort that you have put on our paper. We have carefully read all your comments and made corresponding modifications. Your specific __`comments`__ and our respective responses are noted below. Certain comments are abbreviated or partially quoted for brevity. We hope that the revision addresses all of your concerns.
>
> __`1-“I have not seen any theoretical results on conditional neural networks and I believe this paper is one of the first to tackle theoretical problems for conditional networks; this is a plus. However (perhaps because I'm a theory person), I am not sure if this class of architecture involving explicit if-else branching is really widely used in practice.”`__
>
> At the end of Section 2, we now provide a detailed paragraph on how the if-else model encompasses different practical models in the literature, including early exit or mixture of experts architectures.
>
> __`2-“However, my honest opinion is that both the sufficiency and necessity results are not significant enough to merit acceptance. For sufficiency results, one can easily prove the same statement by constructing a conditional network in the following way: …”`__
>
> Here, the reviewer provides a sketch of a construction to demonstrate the sufficiency of $O(\log n)$ operations for memorization. The reviewer’s sketch will indeed work – we know it because it relies on the same construction that is provided in Theorem 2! Hence, we find the reviewer’s assessment here unfair in the sense that they are using the same arguments that is provided in our current manuscript to claim that the same statement “can easily be proved.”
>
> __`3-“… To me, Theorems 1 and 2 look like a detour to prove a simple thing in an unnecessarily more complicated way.”`__
>
> The reviewer’s observation that the proof for the sufficiency of $O(\log n)$ operations can be simplified is correct. This can be done, e.g. by directly proposing the conditional neural network provided by applying Theorems 1 and 2 in succession. The resulting network would be similar to what the reviewer is suggesting in his second comment. The reason why we did not follow this route of presentation is as follows: The reviewer mentioned that they are a theory person. As a theory person, the reviewer would appreciate that often the means to reach the final result is more important than the final result itself. To us, Theorem 1 is of fundamental importance because it bridges unconditional and conditional networks. In particular, in many existing works on conditional computation, it is implied that sparse feature vectors or representations result in low computational complexity, because zeroes can just be “ignored.” FLOPs are calculated based on the assumption that the feature dimension equals the number of non-zero components. This is, of course, not trivial because if the locations of the zeroes are not structured, then it is not clear how the computational savings can be realized. Theorem 1 shows that one can really ignore the zeroes, at least for the case of ReLU networks.
>
> Another important observation is how Theorem 1 simplifies the design of conditional networks: We no longer have to worry about where to place the gates, for example. All that must be done is to come up with an architecture that produces sparse representations (e.g. using regularization), and the synthesis theorem can automatically exploit the resulting sparsity, in the best possible manner. We thus view Theorem 1 as a major result of the paper, in addition to the $O(\log n)$ necessity-sufficiency results.  To emphasize this point, we have added the following sentence at the end of the abstract: “Our achievability result utilizes a general methodology to synthesize a conditional network out of an unconditional network in a computationally-efficient manner, bridging the gap between unconditional and conditional architectures.” We have also added a new paragraph on the usefulness of Theorem 1 at the end of Section 3 (omitted here for brevity), summarizing the contents of our response to the reviewer.
>
> __`4-“… Theorem 3 could be further improved to capture the right dependence on input/output dimension, as also noted by the authors.”`__
>
> What we had noted was that the gap between Theorems 2 (upper bound) and 3 (lower bound) can potentially be closed. It is not clear to us whether Theorem 3 can be improved further and we actually suspect that the bound may be tight under the very general condition on the dataset provided in the theorem statement. To address the reviewer’s comment, we instead improved Theorem 2, from $O(q(q+r)\log n)$ to $O(q^2 \log n)$, ignoring the constants that do not depend on $n$ (the bounds in the paper provide the constants). We recall that the lower bound provided by Theorem 3 is $\Omega(\log n)$. Hence, we were able to entirely remove the dependence on the output dimension $r$ in the $n$-asymptotic regime.

---

> ### Author Response · Authors · 2022-11-18
> **Response to Reviewer 4ix4 (2 of 3)**
>
> (2 of 3)
>
>
> __`5-“Another weakness is that the paper is not cleanly written, especially the proof part.”`__
>
> We have clarified all proofs as detailed in our responses to the reviewers. In particular, we did a major rewrite of the Proof of Theorem 1 to make it formal, and also formalized the proof of Theorem 2.
>
> __`6-Missing references`__
>
> We have included both references as suggested by the reviewer.
>
> __`7-Expressive power results - Skip connections`__
>
> We have allowed skip connections mainly for the converse results (lower bounds on the required number of neurons, such as in Theorem 3). This is because to obtain the most general lower bound, one should consider the most general architecture. For achievability results, we skip one layer at most, which is a bounded amount. We were unable to identify a network that can achieve the same input-output relationships as in Lemma 1 without layer skipping. If the reviewer is willing to provide the architecture that they believe will accomplish this, we can include it in the final version of the paper. To clarify, we have added the following sentences to Section 2.1: “We also allow skip connections, which are mainly utilized for the generality of the model for converse results (The construction in our achievability results also skip layers, but at most one layer at a time).”
>
> __`8-“The proof of Theorem 1 left me with three questions…”`__
>
> Q1: We originally constructed the tree in the manner described by the reviewer as then, the nested conditionings of the conditional network have a one-to-one correspondence with the unused conditions on the configuration tree. In other words, the constructed conditional network was consistent with the configuration tree. Following reviewer’s comment, we realize our choice made the exposition less clear as opposed to clearer, and thus in the current revision, we only add the output edges that ``exist’’ and correspond to actual inputs.
>
> Q2: This is now addressed in detail at the end of the Proof of Theorem 1 in Appendix A: For the purposes of memorization, it really does not matter what we do with the unused edges, as they are never visited for any member of the training dataset. Nevertheless, one can simply remove binary conditions with one or more empty bins (and “stitch” the ends) to come up with an overall well-defined conditional network graph for any input.
>
> Q3: On the tree, the path that reaches a particular node carried the information of all output configurations at all Layers $0,\ldots,\ell-1$, which were provided as inputs to Layer $\ell$ in the case of skip connections. However, we realize this relationship was not made very clear. In the revised manuscript, we have modified the vertex definitions of the tree to incorporate all output configurations of previous layers. We hope that this clarifies how skip connections are dealt with.
>
> __`9-The number of neurons and layers in Lemma 1`__
>
> Yes, it should read $2q+2$ instead of $q+2$. However, the claim that the number of layers in the construction is $2$ was correct: The first and the hidden layer is formed by the two neurons that generate outputs $y^+$ and $y^-$, and the second or the output layer is formed the $2q$ neurons that generate the outputs $v_j^+,v_j^-,j\in\{1,\ldots,q\}$. To the best of our knowledge (see e.g. (Haykin, 2008) as referenced in the paper), this is the standard way of counting the number of layers.
>
> __`10-“P2: A very useful too -> tool”`__
>
> We have corrected it.
>
> __`11-“P6: In Step-2, what is $d$? If you meant the input dimension it should be $p$.`__
>
> Yes, we meant the input dimension. We corrected it as “$p$.”
>
> __`12-“Thm 2: every neuron in network $f$ has at most $q$ weights—add "incoming" before weights? `__
>
> Added.
>
> __`13-“P7: $n_w$ and $n_e$ are both defined to be "the number of weights of the network," the difference is not at all clear. `__
>
> This was a typo, thank you for pointing it out. The symbol $n_e$ was supposed to be defined as “the number of neurons.” We have corrected it.
>
>
> __`14-“In Def 1, it seems that the authors defined "input" to a neuron to be each of the incoming directed edges. However, this confused me quite a bit because I thought the input to a neuron is just a single scalar, a weighted combination of the output of the previous layer…. Please consider trying to clarify the definition of "input" to a neuron.”`__
>
> We have provided the following clarifying sentences immediately before Definition 1: “We first recall some standard terminology (Haykin, 2008). Consider the neuron $[x_1,\ldots,x_n] \mapsto \phi(\sum_i x_i w_i)$. We refer to $x_1,\ldots,x_n$ as the neuron inputs, and $w_1,\ldots,w_n$ as the neuron weights.”

---

> ### Author Response · Authors · 2022-11-18
> **Response to Reviewer 4ix4 (3 of 3)**
>
> (3 of 3)
>
> __`15-“The definition of depth $\ell$ relative to layer $\ell$ is confusing...”`__
>
> Thank you very much for pointing this out and we apologize for the confusion. In an earlier (pre-ICLR-submission) version of this theorem, the root node began at Layer 1, which made the theorem slightly less general. In the submitted version of the paper, we made the root node begin at “Layer 0” (the input layer), but however forgot to shift certain indices by one unit, which created the problems that the reviewer mentions. We now fixed these typos. In particular, Depth $\ell$ is equivalent to Layer $\ell$, and we corrected incorrect wording in Example 1 accordingly.
>
> __`16-“The definition of root node is also confusing…”`__
>
> The source of this problem is the same as the one described in Comment 15 by the reviewer and is now corrected. The root node indeed corresponds to the input layer.
>
> __`17-“All in all, while the proof delivers the main idea, I thought it might be helpful to provide concrete "algorithms" that construct a configuration tree from an unconditional network and then construct a conditional network from the configuration tree.”`__
>
> The proof of Theorem 1 now describes concrete algorithms: One constructing a configuration tree from an unconditional network, and another constructing a conditional network from the configuration tree, as requested by the reviewer.
>
> __`18-“The paper could use some polishing passes to clean up the confusing definitions too...”`__
>
> We carefully performed multiple passes on the entire paper to clarify the definitions and all proofs.

---

> > ### Comment · Reviewer_4ix4 · 2022-12-02
> > **Thanks for the response**
> >
> > I appreciate the authors for their response, and also the updates to the paper. The response clarified many of the issues I brought up, but please allow me to further comment on a few items.
> >
> > Regarding Item 2, the authors say that it is unfair to just reiterate their main idea of construction and claim it can be easily proved. I agree with the authors that it would be unfair if I didn't devise "my construction" on my own; however, in fact, I came up with the construction I described in the review when I read through the abstract for the first time. At that moment, I wasn't sure of the precise definition of "conditional neural network" just from the description in the abstract, but when I read Section 2 I found that my definition was correct. As I next went through the proofs, I realized that the gist of the construction was exactly the same as what came to my mind in less than a minute. In my subjective opinion, I honestly wouldn't consider a theorem "interesting" if I can correctly guess the main proof idea in a minute.
> >
> > Having said that, I'm kind of convinced by the authors' point on Item 3. While Corollary 1 may have a "more direct" proof that I described, the conversion theorem could be of broader interest, and it could be used in future results that study conditional neural networks. I believe that it would be more interesting if the paper could provide another use case of Theorem 1, but I guess it's a topic for another paper...
> >
> > Re: Item 7, can't we remove the skip connection from $v_j$'s to the input of the 2nd hidden layer by adding $q$ more neurons to the 1st hidden layer to directly "pass" the $v_j$'s? More specifically, one can add, say, $w_j = \phi(v_j + C_j)$ for large enough $C_j \geq 0$'s to shift the input far enough to make sure $w_j = v_j + C_j$, and then define $v_j^+$ and $v_j^-$ using only $w_j$, $y^+$, and $y^-$ (you'll have to "unshift" by $-C_j$ accordingly here). I haven't checked if this idea works, so please see if it does!
> >
> > Minor comment: I checked the newly added citations; Yun et al 2019 uses $O(\sqrt n)$ neurons and $O(n)$ weights, not $O(\sqrt n)$ weights and $O(n)$ neurons.
> >
> > Overall, considering the points raised by the authors and the improvements made in the presentation/clarity, I have decided to raise my score a bit.

---

> > > ### Author Response · Authors · 2022-12-05
> > > **Thank you very much for your reply.**
> > >
> > > We are happy to hear that our revision has clarified many of the issues raised by the reviewer. We appreciate the reviewer's comments, and the fact that the reviewer raised their score.
> > >
> > > In our earlier response, we wanted to clarify that a version of the reviewer's construction already exists in the present paper as a combined result of Theorems 1 and 2. In fact, that is the effect of the synthesis theorem, as it says in particular that there is a one-to-one correspondence between the network in Theorem 2 and what would be its conditional version...  If accepted, we will note in the revised version that a more direct and faster proof that utilizes just the combined construction of Theorems 1 and 2 is possible.
> > >
> > > Regarding the surprise factor, the reviewer would appreciate that even a sentence or two often describing the solution to a research problem conveys a lot of information to the experienced researcher, but it is often difficult to come up with those two sentences in the first place. For example, once one sees a $\log n$ (as in the abstract of the present paper), one may deduce that the construction will very likely be a binary tree, and that the result is "trivial." In fact, every result is trivial once it is known how to achieve it, and sometimes the more difficult part is to find the right model for the problem and the construction to solve it. Of course, these two steps are hidden to the reader, because the paper begins with a polished "model+construction." A natural model, when combined with good writing (!), sometimes will give the reader the impression that everything is trivial. For example, a previous pre-ICLR version of this paper was not using the right model for measuring computational complexity, and we spent a lot of time developing the right model... We ultimately completely respect the reviewer's opinion about the surprise factor, which is completely subjective and very difficult to measure anyway, and once again thank them for shifting their score in the positive direction.
> > >
> > > Regarding the reviewer's comment about item 7, yes we can avoid the skip connection. We do not need to add and remove constants as all the components are all positive due to pre-processing in Layer 1. We will update the manuscript accordingly if accepted. We will also update the reference to Yun et al 2019.

---

> ### Author Response · Authors · 2022-11-28
> **Have we addressed your comments in a satisfactory manner?**
>
> Dear Reviewer 4ix4,
>
> Thank you very much again for your helpful comments and feedback. We were wondering whether we have adequately addressed your questions and concerns. Please let us know if we can answer any more questions or doubts, or any parts that needs further clarification or improvement. Thank you!

---

### Official Review · Reviewer_snry · 2022-10-26

**Confidence:** 2
**Correctness:** 3
**Technical Novelty And Significance:** 3
**Empirical Novelty And Significance:** Not applicable
**Recommendation:** 6

**Clarity, Quality, Novelty And Reproducibility:**

The paper is clearly written and novel to my knowledge.
All the proofs have detailed steps and look reproducible to me.


**Strength And Weaknesses:**

## Strengths:

- The paper is well written and clear
- The result seems significant, especially since conditional neural networks are becoming more mainstream.

## Weaknesses

- The authors don't include the mixture of experts model (Fedus et al. 2021) in their discussion of conditional computation.

**Summary Of The Paper:**

In this paper, the authors focus on neural networks with conditional computation and study the memorization capacity of these networks. The authors show that conditional ReLU networks can memorize n input-output relations in just $O(\log n)$ operations, which is also shown to be the best rate possible.

**Summary Of The Review:**

Overall, the paper is interesting, and very timely and relevant for the community. The results also seem significant. So I vote for accept.

---

> ### Author Response · Authors · 2022-11-18
> **Response to Reviewer snry**
>
> Dear Reviewer,
>
> Thank you very much for your constructive comments and the time and effort that you have put on our paper. We have carefully read all your comments and made corresponding modifications. Your specific __`comment`__ and our respective response are noted below. We hope that the revision addresses all of your concerns.
>
> __`1-“The authors don't include the mixture of experts model (Fedus et al. 2021) in their discussion of conditional computation.”`__
>
> Thank you very much for bringing this very relevant work to our attention. We have added it to our list of references and discuss it at the end of Section 2. This paper also serves as a practical example of the if-else conditional computation model that is considered in the paper, as requested by another reviewer.

---

> ### Author Response · Authors · 2022-11-28
> **Please let us know if you have more questions or concerns**
>
> Dear Reviewer snry, Thank you very much again for your helpful comment that greatly improved the presentation of our work and helped it connect to very important existing literature. We hope that our revision has adequately addressed your concern. Please let us know if we can answer any more questions or doubts, or any parts that needs further clarification or improvement. Thank you!

---

> > ### Comment · Reviewer_snry · 2022-12-12
> > **Thank you for your response.**
> >
> > I would like to thank the authors for their updates, which addressed my specific concern on discussing a recent paper on conditional computation. However, after reading the other reviews, I will keep my score unchanged -- I think the results are interesting since it addresses methods that are used more in deep learning, but I share some of the other concerns with the other reviewers.

---

### Official Review · Reviewer_DoNR · 2022-10-28

**Confidence:** 4
**Correctness:** 3
**Technical Novelty And Significance:** 3
**Empirical Novelty And Significance:** Not applicable
**Recommendation:** 6

**Clarity, Quality, Novelty And Reproducibility:**

**Clarity, Quality, Novelty:**
- The idea and the bound is novel. I think it is a good contribution to a well studied problem.
- The treatment to past literature is excellent although I feel that the proofs could be explained a little better.

**Strength And Weaknesses:**

**Strenghts:**
1. The problem of memorization is well studied and this bound represents an exponential improvement in the minimum number of computations required to memorize a given dataset.
2. The literature survey is thorough and well written, giving proper credit to past work. Since this is a problem with a rich history, it is good to see that it has been given the right treatment.
3. The paper is fairly well written, but the proofs could be better explained.

**Weaknesses:**
1. The previous tight bounds (Rajput et al. for threshold networks, Vardi et al. for ReLU networks) considered the problem of minimizing the number of parameters required for memorization and considered traditional neural networks. The framework of conditional computation, while interesting is not particularly prevalent. So this bound leverages an architecture which may not be practical.
2. The proofs and descriptions in the paper rely a little too much on exmaples. I would encourage the authors to try to find a more formal way of describing ideas such as Conditional Networks in Section 2.3.
3. If I understand correctly, the final architecture is essentially a binary decision tree for every input. This network is quite large ($\mathcal{O}(n \log n)$) parameters and the bit complexity of each weight in it can be arbitrarily large.
    - The prior bounds all need to make some assumption on separation of the data points. Usually something like $||x_i - x_j|| \geq \delta\; \forall i\neq j$. This result requires no such assumption and I suspect this is because the notion of separation is implicitly absorbed in the biases which have infinite precision.
4. The notion of conditional computation breaks down for any kind of batch operation. Most neural networks rely heavily on batch computations because matrix multiplications have been heavily optimized. Such an architecture will not be able to leverage such optimizations and will have to resort to going over the samples serially.
5. Minor:
    - Page 6 has a typo - "configurationtree"
    - Please replicate Figure 4 in the appendix as well to make it easier to go through the proof. I would also encourage you to rewrite Step 3 of the proof to make it easier to parse.

**Summary Of The Paper:**

The paper considers the well studied problem of memorization using deep neural networks from the perspective of the minimum number of computations required per input specifically in the conditional computation framework. They show that $\mathcal{O}(\log{n})$ operations are sufficient to memorize $n$ arbitrary (input, output) pairs using a conditional ReLU network.

**Summary Of The Review:**

I feel there are some minor flaws in the paper and I have a few clarifying questions. If those are handled satisfactorily, I will recommend acceptance.

---

> ### Author Response · Authors · 2022-11-18
> **Response to Reviewer DoNR (1 of 2)**
>
> (1 of 2)
>
> Dear Reviewer,
>
> Thank you very much for your constructive comments and the time and effort that you have put on our paper. We have carefully read all your comments and made corresponding modifications. Your specific __`comments`__ and our respective responses are noted below. Certain comments are abbreviated or partially quoted for brevity. We hope that the revision addresses all of your concerns.
>
> __`1-“The paper is fairly well written, but the proofs could be better explained.”`__
>
> We have performed a major rewrite of proofs, in particular the proofs of Theorems 1 and 2, to explain them better and make them formal.
>
> __`2-“The previous tight bounds (Rajput et al. for threshold networks, Vardi et al. for ReLU networks) considered the problem of minimizing the number of parameters required for memorization and considered traditional neural networks. The framework of conditional computation, while interesting is not particularly prevalent. So this bound leverages an architecture which may not be practical.”`__
>
> Conditional networks have becoming more and more prevalent in addressing many practical machine learning problems recently. In the revised version of the manuscript, we provide several examples at the end of Section 2. In particular, switch transformers (Fedus et al., 2022) have shown tremendous success, achieving significant improvements of inference time relative to an unconditional transformer. Likewise, early exit models can reduce the computational cost without sacrificing model accuracy. We show that both models can be realized as special cases of our general conditional computation model. As such, we believe that the framework of conditional computation as considered in the present work is useful for practical applications and datasets.
>
>
> __`3-“The proofs and descriptions in the paper rely a little too much on exmaples. I would encourage the authors to try to find a more formal way of describing ideas such as Conditional Networks in Section 2.3.”`__
>
> Following reviewer’s suggestion, we now formalize conditional networks as full rooted binary trees, where the edges represent the conditions and the vertices represent the sub neural networks to be executed, as described in the first paragraph of Section 2.3. We have also rewritten the proof of Theorem 1 to a great extent to formalize all steps. We have rewritten parts of the proof of Theorem 2 to also present them in a formal way. We still keep the various examples for a clearer exposition.
>
>
> __`4-“If I understand correctly, the final architecture is essentially a binary decision tree for every input. This network is quite large ($O(n\log n)$) parameters and the bit complexity of each weight in it can be arbitrarily large. The prior bounds all need to make some assumption on separation of the data points. Usually something like $\|x_i - x_j\| \geq \delta;\ \forall i \neq j$. This result requires no such assumption and I suspect this is because the notion of separation is implicitly absorbed in the biases which have infinite precision.”`__
>
> This is correct. The weights and the biases have infinite precision so that it is sufficient if the inputs are distinct. We now discuss this issue in the conclusions in detail: “Extensions of our results to different activation functions or to networks with bounded bit complexity can also be considered. In this context, Vardi et al. (2022) shows that, for every $\epsilon\in[0,\frac{1}{2}]$, $\Theta(n^{\epsilon})$ weights with $\Theta(n^{1-\epsilon})$ bit complexity is optimal for memorizing $n$ patterns, up to logarithmic factors.  This result was proven under a mild separability condition, which restricts distinct dataset patterns to be $\delta$-separated in terms of Euclidean distance. The optimality of the results of Vardi et al. (2022) suggests that under a similar separability condition, the bit complexity of our designs can also potentially be reduced without loss of optimality. This will remain as another interesting direction for future work.”

---

> ### Author Response · Authors · 2022-11-18
> **Response to Reviewer DoNR (2 of 2)**
>
> (2 of 2)
>
> __`5-“The notion of conditional computation breaks down for any kind of batch operation. Most neural networks rely heavily on batch computations because matrix multiplications have been heavily optimized. Such an architecture will not be able to leverage such optimizations and will have to resort to going over the samples serially.”`__
>
> As long as the conditioning tree is not too large, batch operation can still be exploited. For example, assume that there is only one conditioning gate on the network, after which we decide to execute either one of two sub-DNNs. Suppose the input batch size is 1000. We can first pass the entire batch up to the gate to decide on the branches. We can exploit the batch operation that the reviewer mentions in this step. Suppose, after the gate, that 600 samples should travel to one DNN, and 400 samples to the other. These batch sizes are still large enough to benefit from matrix multiplication speed-ups. The different branches can also be processed in parallel, as described in (Fergus et al. 2022). So, networks with mild conditioning will not suffer a big disadvantage in terms of batching. Fortunately, only a few gates can provide significant performance gains, as earlier works show.
>
> For networks with aggressive conditioning, one of the important ideas presented in the current work is to synthesize a conditional network out of an unconditional network. We can then potentially train an unconditional network with sparsity constraints (meanwhile exploiting all the benefits of batching), and then synthesize the conditional network out of the trained unconditional network. In this manner, we bypass the batching problem.
>
> Also, in early exit architectures, the state-of-the-art training method is to pass the entire batch through all the branches (even if a branch will not execute samples of that batch during inference), as shown in (Kaya et al. 2019). So, we “batch anyway.” A similar situation occurs in training neural decision trees with soft gates.
>
> In general, we would agree more research is needed, especially in the aggressive conditioning scenario, and in the inference setting. This problem, however, appears to remain outside the scope of the present work. To address the reviewer’s comment, we added a new paragraph as the second last paragraph in the conclusions section, summarizing all of our response above.
>
> __`6-“Page 6 has a typo - "configurationtree"”`__
>
> We have corrected it.
>
> __`7-“Please replicate Figure 4 in the appendix as well to make it easier to go through the proof. I would also encourage you to rewrite Step 3 of the proof to make it easier to parse.”`__
>
> Following reviewer’s suggestion, we have replicated both Figures 3 and 4 in the appendix (both figures were utilized in the proof). We have also rewritten Step 3 of the proof, where we prove the general case in formal manner.

---

> ### Author Response · Authors · 2022-11-28
> **Have we addressed your comments in a satisfactory manner?**
>
> Dear Reviewer DoNR,
>
> Thank you very much again for your helpful comments and feedback. We were wondering whether we have adequately addressed your questions and concerns. Please let us know if we can answer any more questions or doubts, or any parts that needs further clarification or improvement. Thank you!

---

> > ### Comment · Reviewer_DoNR · 2022-12-02
> > **Thank you for resolving the concerns**
> >
> > I would like to thank the authors for rewriting their proofs to improve readability and addressing all fo my concerns. However, after reading the other reviews, I must agree that the technical contribution of the paper is quite limited. As reviewer 4ix4 mentioned, there exists a simpler construction that achieves the same result.
> > Therefore, I am lowering the score but still recommending acceptance.

---

> > > ### Author Response · Authors · 2022-12-05
> > > **Thank you for your response**
> > >
> > > We would like to thank the reviewer for their response. We are glad to hear that our revisions have addressed all their earlier concerns.
> > >
> > > We would like to invite the reviewer to read our response to Reviewer 4ix4 in detail and the subsequent discussion. The construction that Reviewer 4ix4 mentions is a restatement of the final model that one would obtain once Theorems 1 and 2 are applied in succession. Hence, the construction already exists in the paper, albeit not stated explicitly (We now also see this as potentially a missing aspect of presentation, so we shall state it explicitly in the final version if the paper is accepted). In fact, Theorem 2 has the same tree structure that Reviewer 4ix4 mentions... Also, Theorem 1 is very significant on its own, hence the "perceived detour" we chose for presentation. We argued in our response to Reviewer 4ix4 that it is unfair that a construction that is very similar in the paper is stated to argue that the results of our manuscript is simple or trivial. Reviewer 4ix4 raised their score due to our rebuttal. We are sad to see that, in contrast, the reviewer lowered their score, although we appreciate that they are recommending the acceptance of our work anyway.

---

### Official Review · Reviewer_6Yt2 · 2022-10-28

**Confidence:** 4
**Correctness:** 4
**Technical Novelty And Significance:** 4
**Empirical Novelty And Significance:** Not applicable
**Recommendation:** 8

**Clarity, Quality, Novelty And Reproducibility:**

- Clarity: Overall, the paper was pretty clear, though some important sections could be improved (see weaknesses section above).
- Quality: I believe the work is high-quality and correct. I read the proofs in the main text in detail and did not find errors.
- Novelty: I believe the results are novel, though I’m not very familiar with the related work in this area.
- Reproducibility: The results in this paper are theoretical, and detailed proofs are provided (Theorem 2 is proven in more detail in appendix).


**Strength And Weaknesses:**

Strengths
- In my opinion, the fact that conditional ReLU networks can attain asymptotically better efficiency than unconditional ReLU networks (when memorizing a dataset of size n) is an important theoretical result, with significant practical implications for the design of neural networks. In particular, these results provide strong justification for designing neural networks with conditional branches (e.g., mixtures of experts), in cases where one cares more about inference computation time/energy than about the size of the model.
- The theoretical results/proofs are quite general and elegant.
- Although I found the details in certain sections hard to follow (more comments below), overall I found the main ideas of the paper quite clear, and the results compelling.

Weaknesses:
- The clarity of the paper could be improved. For example:
  1. I found the construction in the proof of theorem 1 hard to follow (I had to read it many times); care should be given to making this a lot clearer, and explaining the “why” in addition to the “how” better. I think this is the most important section to make clearer.
  2. It would be helpful to comment on the constant C > 0 from Theorem 1 (what is its value, and why?).
  3. I think the section connecting VC dimension to memorization capacity in section 4 could be made clearer (it also has a typo: In the last sentence of that paragraph, n_e is twice described as the number of weights instead of as the number of neurons).
There are currently no empirical results in this paper.



**Summary Of The Paper:**

This paper proves that conditional ReLU networks can memorize any dataset of size n, in such a way that performing inference with this network requires only $O(\log{n})$ operations per input. It also proves that this is the best possible result, assuming mild conditions on the dataset. The best known results for unconditional ReLU networks, on the other hand, are that datasets of size n can be memorized with $O(\sqrt{n})$ weights and $O(\sqrt{n})$ neurons, with the best possible result being $O(\sqrt{n})$ weights and $O(n^{1/4})$ neurons. Thus, the results in this paper prove the almost exponential improvement in inference efficiency that can be attained by using conditional computation.

The constructions in this paper define conditional ReLU networks with $O(n)$ neurons and $O(n)$ weights. Thus, the paper identifies an important open question for future work: can the $O(\log{n})$ inference time result can be attained with asymptotically fewer neurons/weights (e.g., $O(\sqrt{n})$)?


**Summary Of The Review:**

This paper proves important theoretical results about the efficiency with which conditional ReLU networks can memorize datasets compared with unconditional ReLU networks. These results have important practical implications for the design of neural networks. As a result, I recommend acceptance for this paper (although my review is only medium confidence, given that I am not super familiar with the related work).

---

> ### Author Response · Authors · 2022-11-18
> **Response to Reviewer 6Yt2**
>
> Dear Reviewer,
>
> Thank you very much for your constructive comments and the time and effort that you have put on our paper. We have carefully read all your comments and made corresponding modifications. Your specific __`comments`__ and our respective responses are noted below. Certain comments are abbreviated or partially quoted for brevity. We hope that the revision addresses all of your concerns.
>
> __`1-“... In particular, these results provide strong justification for designing neural networks with conditional branches (e.g., mixtures of experts), in cases where one cares more about inference computation time/energy than about the size of the model.”`__
>
> Thank you for your comment. In fact, we had omitted to cite and mention mixture of experts models. We now provide connections to these works at the end of Section 2.
>
> __`2-“The clarity of the paper could be improved. For example: ...”`__
>
> We have performed a major revision of the paper to improve clarity, including rewriting most of the proofs and proving more discussions, as described in our response to reviewer(s).
>
> __`3-“I found the construction in the proof of theorem 1 hard to follow (I had to read it many times); care should be given to making this a lot clearer, and explaining the “why” in addition to the “how” better. I think this is the most important section to make clearer.”`__
>
> We have rewritten the proof to make it clearer and more formal. The proof is now provided in Appendix A. We provided motivations on why we provide certain constructions.
>
> __`4-“It would be helpful to comment on the constant C > 0 from Theorem 1 (what is its value, and why?).”`__
>
> We now calculate the computational complexity explicitly, and the $C \omega_n$ upper bound is replaced with the precise bound $p + 4\omega_n$. So, under the assumption that the number of active weights is larger than the input dimension (which happens for most practical networks), one can take $C=5$. Actually, as we also discuss after the theorem statement, “The proof of the theorem suggests that the true complexity is closer to $2 \omega_i$ than the actual formal upper bound provided in the theorem statement. The number $2 \omega_i$ stems from the $\omega_i$ additions and $\omega_i$ multiplications that are necessary to calculate the local fields of active neurons.”
>
> __`5-“I think the section connecting VC dimension to memorization capacity in section 4 could be made clearer (it also has a typo: In the last sentence of that paragraph, n_e is twice described as the number of weights instead of as the number of neurons). There are currently no empirical results in this paper.”`__
>
> We have corrected the typo and extended the discussion to make it clearer. We have addressed the question of why the current work does not provide empirical results in the second paragraph of Section 1.3. The focus of the present work is on fundamental limits, and we believe the results are precise enough to show that conditional networks can offer significant benefits over unconditional networks. The presented constructions are specifically tailored to extract the fundamental gains in the asymptotic limit $n\rightarrow\infty$ of an infinitely large dataset. Applying such constructions to practical datasets of finite cardinality may result in suboptimal performance. Training for a practical problem also requires generalization, which is little understood not only in the context of memorization but also in conditional computation. We thus believe an empirical study is a significant undertaking by itself and thus remains beyond the scope of the present work.

---

> ### Author Response · Authors · 2022-11-28
> **Have we addressed your comments in a satisfactory manner?**
>
> Dear Reviewer 6Yt2,
>
> Thank you very much again for your helpful comments and feedback. We were wondering whether we have adequately addressed your questions and concerns. Please let us know if we can answer any more questions or doubts, or any parts that needs further clarification or improvement. Thank you!

---

### Decision · Program_Chairs · 2023-01-20

**Decision:**

Accept: poster

**Justification For Why Not Higher Score:**

Paper has limited technical contribution

**Justification For Why Not Lower Score:**

Paper introduces a new framework to study conditional compute architectures.

**Metareview: Summary, Strengths And Weaknesses:**

This paper extends existing memorization results for feedforward networks to conditional computation models. It presents a clean framework and results for the memorization capacity these models. The main concerns were around limited technical novelty and contributions, however all the reviewers agreed that the introducing this novel framework can be of interest to broader community given the increasing interest in architectures such as Mixture of experts. Hence I suggest acceptance. I strongly encourage authors to update paper following reviewers suggestions - in particular finding more applications of the framework can strengthen the paper significantly.

Pros -
1. Novel framework to study conditional computation architectures.
2. Results are well presented with clear proofs.

Cons -
1. Limited technical novelty and contributions.

**Note From Pc:**

if the above contains the word "oral" or "spotlight" please see: "oral" presentation means -> notable-top-5% and "spotlight" means -> notable-top-25%. As stated in our emails, we are disassociating presentation type from AC recommendations

**Summary Of Ac-Reviewer Meeting:**

The main concern brought up during the discussion was about limited technical novelty in the paper as the key results follow from simple arguments using existing results for feedforward networks. However the paper has merits in introducing a general framework to think about architectures with conditional compute. Reviewers felt this is a nice contribution as there is growing interest in such architectures and the paper can be a good step towards understanding such models better.